# Ionization and structural properties of mRNA lipid nanoparticles influence expression in intramuscular and intravascular administration

Manuel J. Carrasco[1], Suman Alishetty[1], Mohamad-Gabriel Alameh[2], Hooda Said[1], Lacey Wright[1], Mikell Paige[3], Ousamah Soliman [2], Drew Weissman[4], Thomas E. Cleveland IV [5], Alexander Grishaev[5] & Michael D. Buschmann [1✉]

Lipid Nanoparticles (LNPs) are used to deliver siRNA and COVID-19 mRNA vaccines. The main factor known to determine their delivery efficiency is the pKa of the LNP containing an ionizable lipid. Herein, we report a method that can predict the LNP pKa from the structure of the ionizable lipid. We used theoretical, NMR, fluorescent-dye binding, and electrophoretic mobility methods to comprehensively measure protonation of both the ionizable lipid and the formulated LNP. The pKa of the ionizable lipid was 2-3 units higher than the pKa of the LNP primarily due to proton solvation energy differences between the LNP and aqueous medium. We exploited these results to explain a wide range of delivery efficiencies in vitro and in vivo for intramuscular (IM) and intravascular (IV) administration of different ionizable lipids at escalating ionizable lipid-to-mRNA ratios in the LNP. In addition, we determined that more negatively charged LNPs exhibit higher off-target systemic expression of mRNA in the liver following IM administration. This undesirable systemic off-target expression of mRNA-LNP vaccines could be minimized through appropriate design of the ionizable lipid and LNP.

[1] Department of Bioengineering, George Mason University, Fairfax, VA, USA. [2] Perelman School of Medicine, University of Pennsylvania, Philadelphia, PA, USA. [3] Department of Chemistry & Biochemistry, George Mason University, Fairfax, VA, USA. [4] Perelman School of Medicine, University of Pennsylvania, Philadelphia, PA, USA. [5] Institute for Bioscience and Biotechnology Research National Institute of Standards and Technology, Rockville, MD, USA. ✉email: mbuschma@gmu.edu

essenger RNA (mRNA) vaccines are now accepted modalities following the Emergency Use Approval of two highly efficacious vaccines that are based on lipid nanoparticle (LNP) delivery of nucleoside-modified mRNA sequences encoding for a modified version of the SARS-CoV-2 spike protein[1,2]. mRNA LNPs bear a structural resemblance to viral systems and circulating endogenous lipid-containing chylomicrons in terms of size and lipid envelope[3]. These features contribute to the application of mRNA-LNPs as delivery vehicles for vaccines and other therapeutics including transcripts encoding antibodies for endogenous translation[4] or enzymes for in vivo gene editing[5]. It is noteworthy that all 8 mRNA vaccines in clinical trials for SARS-CoV-2 use LNPs as the delivery system.

The performance of LNPs for mRNA delivery depends on their potency[6], in vivo distribution and targeting[7], induction of inflammatory and innate immune reactions[8], and for vaccines, their inherent adjuvanticity[9]. Potency, which refers to delivery efficiency, has been the central focus of most LNP research culminating in the commercial approval of the Onpattro™ siRNA containing LNP for liver-based gene silencing[10]. The central feature of the ionizable lipid that was necessary for high delivery efficiency was the pKa of the LNP needed to be between 6 and 7, as determined using the anionic fluorescent dye 2-(p-toluidino)-6-naphthalenesulfonic acid (TNS) binding assay. With a pKa near 6.5, the LNP at neutral pH is negatively charged (from the nucleic acid payload). However, once inside the cell in endosomes the pH declines to near 4.5 prior to lysosomal fusion thereby protonating the LNP, which facilitates ion pair formation of the cationic ionizable lipid with anionic endogenous endosomal phospholipids[3]. At this point, the second feature that is known to control potency comes into play, namely the molecular shape of the ionizable lipid[11,12]. A more cone-shaped (versus cylindrical) ionizable lipid is desirable since a small head group and broader molecular cross-section in the tail region are incompatible with a lipid bilayer thereby destabilizing the endosomal membrane and allowing the nucleic acid payload to be released to cytosol. The specific ionizable lipids in the approved mRNA-LNP vaccines have an accentuated molecular cone-shape through increased lipid tail branching for Lipid H[13] (SM-102)[14] from Moderna, and for ALC-0315[15] from Acuitas in the BioNTech/Pfizer vaccine.

LNP electric charge is known to control in vivo distribution and expression of mRNA-LNPs for intravascular (IV) administration. Reducing the amount of cationic lipid in mRNA-LNPs created a negative LNP, due to an excess of anionic mRNA, that targeted the spleen after IV injection, versus positively charged LNPs targeting the lung[16]. Similar charge-mediated targeting with IV delivery was also achieved using standard LNPs and mixing in a permanently cationic lipid or a permanently anionic lipid to endow the LNPs with a net positive, net negative, or an intermediate net charge that targeted the lungs, spleen, and liver, respectively[7]. Although charge-mediated targeting has not been examined for intramuscular (IM) administration, many studies that analyze expression after IM injection do find a systemic trafficking of mRNA-LNPs that strongly express in the liver very quickly, while simultaneously expressing in muscle and draining lymph nodes[17–19]. These particular LNPs apparently enter the vasculature after IM injection and subsequently express in liver hepatocytes possibly due to passive ApoE-mediated targeting[20]. Systemic mRNA-LNP distribution and off target expression of immunogens could however generate systemic cytokines, activate complement, amplify the frequency or severity of adverse events that have been observed in recent clinical trials[21,22], and/or impair immune response generation[23]. Only one prior study has been published for the optimization of the ionizable lipid in mRNA-LNPs for IM administration of vaccines, focusing mainly on increasing the degradability of the lipid to improve tolerability[13].

Given the lack of studies that address LNP design principles for IM administration that could lead to a vaccine-optimized mRNA-LNP, we developed several new theoretical and experimental methods that deepen our understanding of the ionization of the ionizable lipid and of the LNP. By calculating theoretical pKa values of the ionizable lipids and measuring them directly with NMR, we found their pKa's to be 2–3 units higher than literature reports of LNP pKa's measured using TNS binding assay. We were able to quantitatively explain this difference through thermodynamic modeling of the ionization equilibria in the LNP versus an aqueous phase, which will enable rational design of LNPs that targets specific ionization requirements based on the structure of the ionizable lipid, thereby streamlining the synthesis and screening of ionizable lipid candidates. We applied these and additional new methods to relate ionizable lipid ionization and LNP ionization to potency in vitro and in vivo for IM and IV administration We further propose design principles to limit off-target systemic distribution and expression for mRNA LNP vaccines and to rapidly design new ionizable lipids that could have both high potency as well as localized expression for IM administration of mRNA LNP vaccines.

## Results

**Molecular ionization properties of ionizable lipids can be predicted theoretically and are 2–3 units higher than the pKa of the corresponding Lipid Nanoparticle formulation.** With the eventual aim of designing ionizable lipids that give rise to the appropriate ionization properties of the LNP, such as its pKa being within the 6–7 range and a net charge that drives the desired targeting, we calculated the theoretical pKa of the following five published and commercially available ionizable lipids: DLin-KC2-DMA, DLin-MC3-DMA, DLin-DMA, DODMA, and DODAP using commercial software and found them to be in the range of 8–9.4 (Table 1). These theoretical pKa's were significantly higher than the previously reported experimental TNS binding assay pKa's (5.8–7)[24–26] of the LNPs containing the same ionizable lipids. We therefore decided to directly measure the ionizable lipid pKa using a well-established NMR method based on the pH dependence of the $^{1}$H chemical shift of protons near the ionizable group[27]. Since the NMR method requires water-soluble compounds, we synthesized water-soluble analogs of these five ionizable lipids, which did not change their theoretical pKa's, and measured the pH dependence of the terminal protons of the dimethylamine in a series of buffers with pH ranging from ~6-12. The NMR pH indicators imidazole and piperazine were included as internal standards to provide the pH of each solution based on the chemical shifts of the protonated nitrogen atom in these reagents (Supplementary Fig. 1) By fitting the chemical shifts of the terminal dimethylamine protons of the analogs to the Henderson–Hasselbalch equation (Fig. 1a), we found the NMR-determined pKa values to agree with the theoretically calculated pKa's (Table 1).

We then made mRNA-LNPs using firefly luciferase (FLuc) nucleoside-modified mRNA and each of the ionizable lipids using mole ratios of ionizable-lipid/DSPC/cholesterol/PEG-lipid of 50/10/38.5/1.5 and an NP lipid:mRNA ratio of 4 for the amine group of ionizable lipid to the phosphate groups of mRNA. The TNS dye binding assay provided pKa's that were similar to those in the literature (Fig. 1b), and we verified that differences in the TNS binding assay conditions found in the literature did not significantly influence the measured TNS binding assay pKa's. (Supplementary Tables 1 and 2 and Supplementary Figs. 2 and 3).

**Table 1 Ionizable lipid structure, pKa by ACD (theory), NMR, TNS, ZP, and pI by ZP.**

| Lipid | Structure | pKa, ACD | pKa, NMR | pKa, TNS | pKa, ZP | n, ZP | pI, ZP | ΔpKa NMR-TNS | ΔpKa NMR -ZP | ΔpKa TNS -ZP |
|---|---|---|---|---|---|---|---|---|---|---|
| DLin-KC2-DMA |  H₂O soluble analogue : | 9.32 | 9.34 | 7.03 | 6.30 | 2.26 | 6.61 | 2.31 | 3.04 | 0.73 |
| DLin-MC3-DMA |  H₂O soluble analogue : | 9.37 | 9.47 | 6.57 | 5.77 | 1.69 | 5.93 | 2.9 | 3.7 | 0.80 |
| DLin-DMA |  H₂O soluble analogue : | 8.65 | 8.93 | 6.66 | 6.15 | 2.52 | 6.34 | 2.27 | 2.78 | 0.51 |
| DODMA |  H₂O soluble analogue : | 8.65 | 8.93 | 6.59 | 5.83 | 3.83 | 6.24 | 2.34 | 3.1 | 0.76 |
| DODAP |  H₂O soluble analogue : | 8.02 | 7.65 | 5.62 | 5.59 | 1.95 | 5.39 | 2.03 | 2.06 | 0.03 |

As a complementary method to the TNS binding assay for obtaining LNP ionization properties, we performed electrophoretic mobility measurements of the LNPs that provide zeta potentials (ZPs) and found that all LNPs carried a positive net charge at low pH and transitioned through an isoelectric point (pI 5.4–6.6) to become negatively charged at high pH (Fig. 1c). In comparing ZP titration curves to TNS binding titration curves, it became apparent that ZP titration occurred over a much broader pH range (~4 points) than the TNS binding titration (~2 points) suggesting that TNS only detects LNP surface charge while ZP detects LNP net charge. This broader titration curve for ZP was reminiscent of the titration of polyelectrolytes where an extended Henderson–Hasselbalch equation is used to fit this behavior and provide ZP p$K$a (Fig. 1c). This stretched titration curve is due to the intrinsic p$K$a of the LNP depending on its ionization state as noted with polyelectrolytes[28,29]. In comparing the NMR p$K$a (that agrees with the theoretical pKa) to the TNS binding assay pKa, we found a consistent 2–3 unit drop and a further ~0.7 unit drop point from the TNS binding assay p$K$a to the ZP p$K$a (Table 1 and Fig. 1d). The latter can be easily understood since TNS binding assay only detects initial surface LNP protonation down to ~pH = 6 (Fig. 1b), while ZP detects net LNP charge and protonation down to pH = 3 (Fig. 1c), which covers the entire pH range in endosomal and lysosomal compartments. The explanation for the 2–3 point drop from NMR to TNS binding assay or ZP p$K$a was not however immediately obvious.

We developed a thermodynamic model that accounts for the measured p$K$a difference between NMR and TNS binding assay (or ZP) p$K$a as detailed in Supplementary Information. By considering the ionization equilibrium of the dimethylamine moiety inside the lipid phase and the equilibrium of protons between the lipid phase and the water phase where the pH is measured during the TNS binding and ZP assays, we derived the following expression relating the measured apparent p$K$a of the LNP, p$K_{appL}$, to the intrinsic pKa of the ionizable lipid in the LNP, p$K_{aL}$, and the free energy of transfer of protons from water to the LNP due to proton solvation, $\Delta G_{tr}(H^+_{W \to L})$, and the electrostatic repulsion or attraction of protons to the LNP due to its electric potential $\Psi_L$. (see Supplementary Information for derivation).

$$pK_{appL} = pK_{aL} - \frac{\Delta G_{tr}(H^+_{W \to L})}{2.303RT} - \frac{F\Psi_L}{2.303RT}$$

This equation results from the fact that the proton concentration in the LNP is not measurable and that when pH is measured in the water phase, this creates the two extra terms on the right in the above equation. As detailed in the Supplementary Information, the high energy of solvation of protons in the lipid versus water phase, $\Delta G_{tr}(H^+_{W \to L}) \cong 15 \to 20 kJ/mol$, reduces the apparent LNP pKa by ~3–4 units while the intrinsic p$K$a in the LNP, p$K_{aL}$, is somewhat higher than in the aqueous phase by ~1 unit, resulting in a net drop of 2–3 units in pKa from the aqueous phase to the LNP phase for the ionizable lipid, which corresponds to the p$K$a drop seen from NMR and theory (aqueous phase) to TNS binding assay and ZP (LNP phase). Proton solvation energy differences are therefore principally responsible for the measured 2–3 unit difference between aqueous molecular NMR/ACD p$K$a measurements vs LNP TNS/ZP p$K$a measurements (Table 1). The electrostatic potential of the LNP, $\Psi_L$, further modulates the apparent p$K$a in an ionization-state-dependent manner shown in the above equation, reducing the pKa at low pH when the LNP is positive and increasing the pKa at high pH where the LNP is negative. The latter effect is also responsible for broadening the ZP titration of the LNP to cover ~4 pH units rather than ~2 units, which is the case for a monoprotic species whereas the assembled

LNP behaves like a polyelectrolyte. This model and the corresponding explanation can now be used for the rational design of new ionizable lipids by theoretically calculating the lipid ionization properties, verifying/correcting those properties by NMR, and then estimating the corresponding LNP ionization properties for the mRNA-LNP systems. For example, novel head groups with both linear and cyclic structures and with different numbers of carbon atoms separating them from the linker can be theoretically analyzed for ionization properties to assess suitability for specific applications.

**Ultrastructural analyses reveal lower levels of C18 lipid saturation create more heterogenous and irregularly structured LNPs.** In addition to ionization and morphology of the ionizable lipid, the ultrastructure of the mRNA-LNP is known to influence potency[30,31]. CryoTEM of LNPs with dilinoleic (C18 - 2 double bonds) tails (KC2, MC3, DLin) showed relatively homogeneous spherically shaped LNPs with an exterior bilayer and electron-dense internal amorphous structure (Fig. 2a–c). This appearance is consistent with a previous model supported by neutron scattering data where the peripheral shell is the polyethylene glycol lipid and is rich in DSPC while the internal compartment is primarily the ionizable lipid electrostatically bound to the mRNA with cholesterol distributed throughout[30]. For the dioleic LNPs, where each lipid tail has only one double bond (C18 – 1 double bond) (DODMA, DODAP), cryoTEM showed heterogenous LNP populations (Fig. 2d–g). DODMA LNPs had a very high frequency of small liposomal structures (Fig. 2e). DODAP had a high frequency of larger segmented multicompartmental structures where an electron-dense amorphous compartment was adjacent to a more electron-lucent compartment with greater internal structure († in Fig. 2f, g) and a peripheral bilayer (Fig. 2f, g). This type of mRNA-LNP structure has been observed previously and the electron-lucent compartment with greater internal structure († in Fig. 2f, g) has been identified by nanogold labeling to contain the mRNA[32]. This multicompartmental structure may result from incomplete fusion of incipient LNPs during dialysis that is an important part of LNP assembly[33]. A recent study[34] found that the population of LNPs prior to dialysis contains both electron-dense and electron-lucent nanoparticles and that as pH is raised to 7.4 during dialysis, the neutralization of ionizable lipid sequesters it from aqueous domains driving the fusion of several vesicles to produce the final electron-dense core surrounded by a bilayer described above for KC2, MC3, and DLin. Apparently, this process was inhibited for DODMA, which retained many small structures. For DODAP, its lower TNS binding assay p$K$a (5.6) may result in a high proportion of neutralized ionizable lipid in the incipient LNPs that then inhibits fusion and LNP reorganization to generate multicompartmental structures with both electron-dense and more electron-lucent zones, the latter containing the mRNA. In addition it is important to note that previous observations concerning the influence of lipid tail saturation on LNP potency have focused on endosomolytic capacity as related to a cone-shaped lipid morphology[26] while a second equally important effect observed here may be the influence of lipid saturation on the fusion process during dialysis and resultant LNP structures. SAXS analyses revealed a peak for the scattering vector (q) near 1 nm$^{-1}$ for each LNP corresponding to a spacing (d) between mRNA phosphate backbone chains in the core of the LNP of near 6.5 nm for all LNPs, in agreement with prior studies[30,33], except for DODAP, which was larger at 7.4 nm. The larger size for DODAP-based LNPs is possibly related to what appears to be a more aqueous multi-compartmental environment for the mRNA.

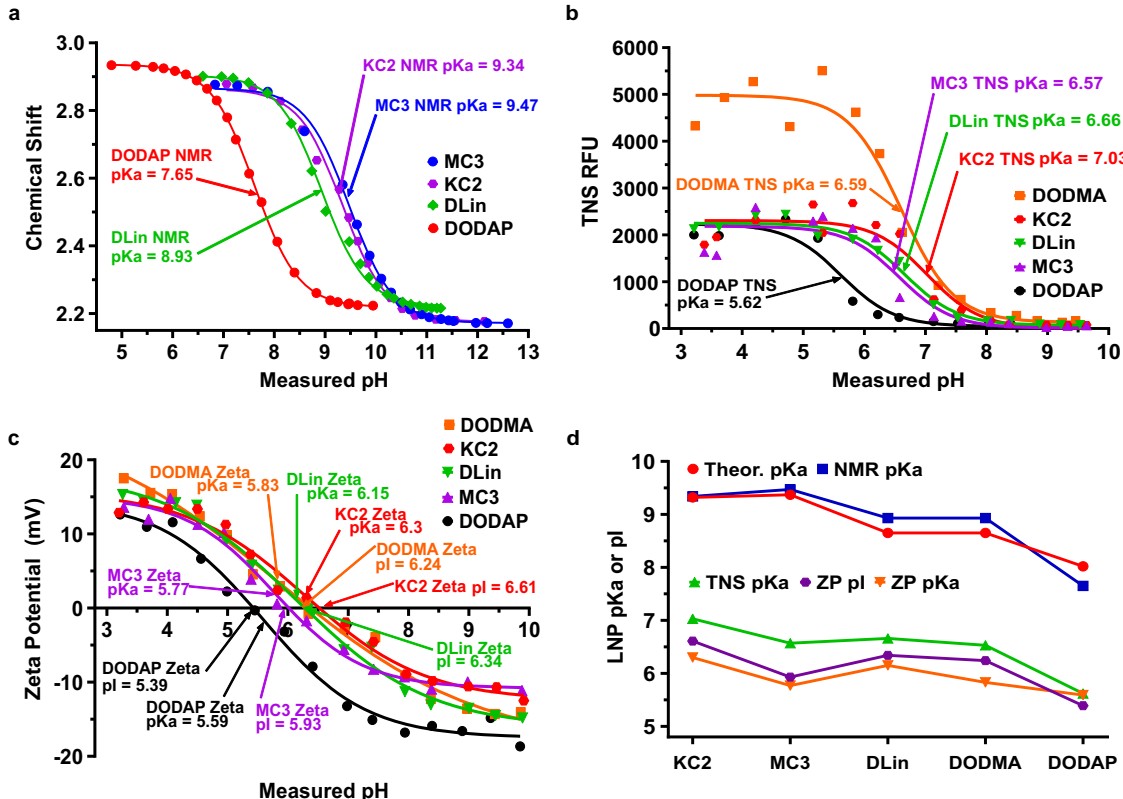

**Fig. 1 Molecular ionization properties of commercially available ionizable lipids DLin-KC2-DMA, DLin-MC3-DMA, DLin-DMA, DODMA, DODAP, and their mRNA LNPs.** mRNA LNPs were assembled with the mole ratios 50:10:38.5:1.5 (ionizable lipid:DSPC:cholesterol:PEG-DMG2000) at NP lipid:mRNA ratio 4. **a** NMR measurement of pKa of custom-synthesized water-soluble analogs of ionizable lipids, fitting the pH dependence of the terminal methyl protons of the head group to the Henderson–Hasselbalch (HH) equation (spectra are in Supplementary Fig. 1). **b** TNS measurement of LNP pKa only detects initial protonation above pH ~6 and fits the HH model to provide TNS binding assay pKa of the LNP. **c** Zeta Potential (ZP) measurement of LNP measured over the pH range 3–10 revealed a transition from a positively charged LNP at low pH to a negatively charged LNP at high pH and detects protonation down to pH 3 covering the endosomal and lysosomal pH range. ZP fits the extended HH model to provide the ZP pKa. pI is the pH where LNP zeta potential interpolates to zero. **d** pKa of the ionizable lipid measured by NMR agrees with theoretical predictions while pKa in the LNP whether by TNS or Zeta Potential were 2–3 points lower.

**In vivo potency of mRNA Lipid Nanoparticles correlates with in vitro potency for intramuscular administration but not for intravascular administration due to LNP charge-mediated trafficking.** In order to understand the influence of the ionizable lipid, LNP ionization properties, and resultant LNP ultrastructure on mRNA-LNP potency, HEK293 cells were transfected with LNPs containing an mRNA encoding a luciferase reporter using a series of increasing doses from 25 to 200 ng of mRNA per well containing 12,000 cells. HEK293 cells were chosen as a first model cell type for potency screening that could be replaced in future studies by primary cells more representative of in vivo targets. We found a very reproducible order of potency for in vitro transfection from high to low delivery and expression efficiency as follows: KC2 > MC3 > DLin>DODMA > DODAP (Fig. 3). DODAP LNPs had almost undetectable expression consistent with a TNS binding assay pKa of 5.62, which is too low for endosomal protonation, although expression with this pKa has been observed in LNPs targeting the spleen[35]. DODAP and DODMA also have heterogeneous ultrastructural features as described above that are expected to reduce potency. DLin has a pKa very similar to MC3 and identical lipid tails but with two ether linkers versus one ester for MC3, which could render DLin more cylindrical and less endosomolytic (less cone-shaped)[36]. A surprising finding was a consistently superior potency for KC2 versus MC3 in vitro in light of MC3 having demonstrated superiority for hepatocyte gene silencing by IV administration[6,24].

One difference between these two LNPs is their charge at physiological pH where KC2 is less negative than MC3 as reflected by a ZP pI of 6.6 for KC2 vs 5.9 for MC3 (Table 1). It is well known that in vitro transfection is generally facilitated by more positively charged or less negatively charged nanoparticles[37] possibly due to negative heparin sulfate proteoglycans coating the cell membrane[38].

When we administered these LNPs into the muscle of Balb/c mice, we found an identical order of potency for luciferase expression consistent with our in vitro experiments (Fig. 4a, c). However, when we intravenously administered the LNPs, the superiority of MC3 to KC2 was established for expression in the liver (Fig. 4b, d) as observed for siRNA delivery[6,10,24]. In vivo potency for IM correlated with in vitro potency with an $R^2 = 99.8\%$ ($p < 0.0001$), while in the liver for IV expression there was no correlation with in vitro potency with an $R^2 = 44.6\%$ ($p = 0.22$) (Fig. 4f). The superiority for KC2 in vitro and for in vivo IM suggests a less negatively charged LNP is more efficient in vitro and in vivo IM, while for IV administration a more negative LNP that is passively targeted through Apo-E absorption is more efficient for hepatocyte targeting. The potential role of negatively charged proteoglycans in the extracellular matrix in muscle, which are absent in blood but present in vitro on cell membranes is consistent with less negatively charged LNPs being more potent for IM administration while the more negative LNPs more quickly pass into the vasculature. Passive Apo-E targeting in blood is also

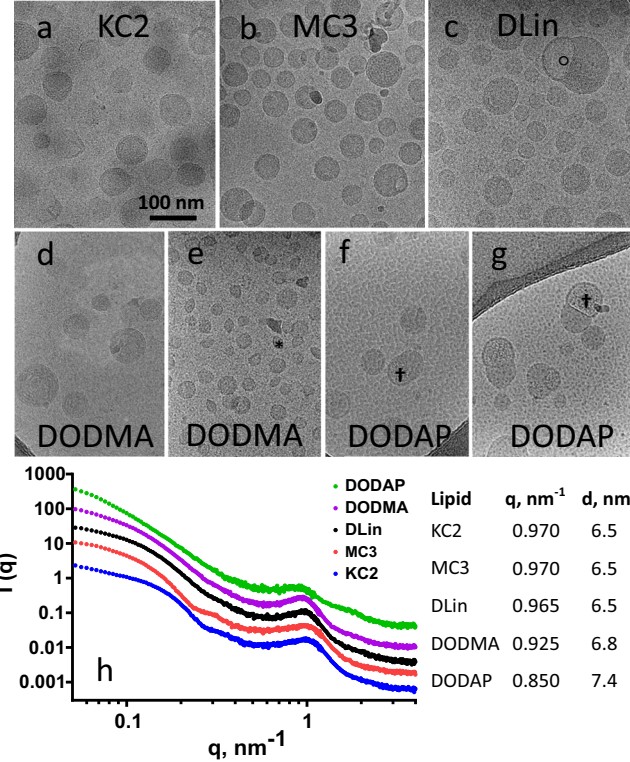

**Fig. 2 Ultrastructural characterization of mRNA LNPs.** CryoTEM of LNPs made with ionizable lipids bearing dilinoleic acid tails (KC2, MC3, and DLin) had a relatively homogeneous appearance (**a**, **b**, **c**) with an evident superficial bilayer and an amorphous electron dense core and occasional electron-lucent cavities (o in **c**). LNPs made from ionizable lipids with dioleic acid tails, each tail having only a single double bond (DODMA and DODAP), displayed heterogeneous ultrastructure (**d**, **e**, **f**, **g**). DODMA LNPs contained a significant population of small liposomal structures (* in **e**). DODAP LNPs were predominantly multicompartmental where an electron-dense amorphous compartment was adjacent to a more electron-lucent compartment with greater internal structure containing the mRNA († in **f** and **g**) and a peripheral bilayer. **h** SAXS analyses revealed internal spacing between mRNA backbones calculated from the peak position at ~1 $nm^{-1}$ and was similar for (KC2, MC3, DLin), increased slightly for DODMA and more so for DODAP. The latter was consistent with the likely presence of mRNA in a compartment with higher water content (†).

known to occur in a charge-dependent fashion since permanently cationic LNPs[20] as well as highly negatively charged LNPs[39] do not exhibit Apo-E binding and therefore do not express in hepatocytes. We also detected the well-known off-target expression in liver for these LNPs injected IM (Fig. 4a). Here, we additionally observed that MC3 with its greater negative charge had a fraction of photon flux in liver (Fig. 4e) that was on average higher than for KC2 suggesting negative LNP charge was facilitating off-target expression after IM administration and is accentuated by a more negatively charged LNP.

**Reducing the NP lipid:mRNA ratio creates larger more negatively charged LNPs that have higher potency due to increased protonation in the endosomal pH range.** To further explore the role that LNP charge could have on potency and targeting, we made KC2 mRNA LNPs with different NP ratios of ionizable lipid amine to mRNA phosphate in the range of 2–8. We found that reducing the NP lipid:mRNA ratio from 8 to 2 increased the LNP diameter from 48 to 72 nm and reduced encapsulation efficiency from 80 to 40%. (Fig. 5a, b). We developed a

quantitative estimate of the number of copies of mRNA in each LNP using the number-averaged diameter of the LNP measured by DLS to calculate the LNP volume and then filling up that volume with each of the 5 components (4 lipids and mRNA), while respecting their mole ratios and assigning their molecular volume to each molecule in the LNP (see Supplementary Information for details). The model also accounts for the encapsulation efficiency by reducing the mole ratio of mRNA in the LNP by the encapsulation efficiency. We estimated that the average number of mRNA copies per LNP increases from 1 at an NP lipid:mRNA ratio of 8 to 6 at an NP lipid:mRNA ratio of 2 (Fig. 5c) due to a 3-fold increase in volume combined with a 2-fold reduction of the NP lipid:mRNA ratio in the LNP (NP ÷ Encapsulation Efficiency) the latter increasing mRNA mole ratio in the larger LNP by 2-fold. By performing the TNS binding assay at a constant ionizable lipid concentration (accounting for the NP lipid:mRNA ratio), we observed that lower LNPs with lower NP lipid:mRNA ratios had higher TNS binding assay fluorescence at low pH suggesting greater surface protonation (Fig. 5d). The ZP measurements showed that reducing NP lipid:mRNA ratio created more negatively charged LNPs where the pI = 7.1 at an NP lipid:mRNA ratio of 8 and pI = 5.5 at an NP lipid:mRNA ratio of 2, while the pKa's did not change as dramatically (Fig. 5f). We additionally calculated the absolute elemental charge of the LNPs (Fig. 5g) using a non-linear electrokinetic model (see Supplementary Information) and were able to estimate the LNP dielectric constant (Fig. 5h) by comparing this measured charge at high pH to the expected charge from the calculated number of mRNA copies in the LNP, since the ionizable lipid is uncharged at high pH. The LNP dielectric constant ranged from 6 to 24, which is intermediate between that of lipid[40] (5) and water (80), thereby supporting the accuracy of the mRNA copy number calculation. Interestingly, the water content of an LNP with an NP lipid: mRNA ratio of 4 was estimated at 25%[30] so that the LNPs from formulations with a high NP lipid:mRNA ratio of 8 could be similar to a pure lipid phase solution with a dielectric constant of 6. The LNPs formulated with low NP lipid:mRNA ratio, and therefore higher mRNA content, would be expected to contain more water and therefore a higher dielectric constant (Fig. 5h). Luciferase expression at a 200 ng dose was found to increase for LNPs made with low NP lipid:mRNA ratios (Fig. 5i). We found this high dose in vitro potency to correlate strongly and negatively with the NP lipid:mRNA ratio (Fig. 6a) and encapsulation efficiency (Fig. 6b), and to correlate strongly and positively with LNP size (Fig. 6c), mRNA copies in each LNP (Fig. 6d), and with the increase in protonation from pH 7.4 to 5 seen in both TNS binding assay and in LNP net charge calculated from the ZP (Fig. 6e, f). The negative correlation seen with encapsulation efficiency may seem surprising, however this may be a consequence of higher mRNA copy numbers per LNP at low NP lipid:mRNA ratios and increased LNP protonation in the endosomal pH range for lower NP lipid:mRNA ratios. Therefore, maximization of encapsulation efficiency may not always be desirable if important potency properties involving protonation are negatively impacted. To summarize these in vitro findings, we found low NP LNPs are more potent, larger, more negatively charged and exhibit larger levels of protonation in the endosomal pH range, which may be one mechanism for increasing potency in our in vitro administration. The latter is consistent with a recent report where the absolute TNS binding assay fluorescence at pH = 5 was strongly predictive of potency[41].

**Characterization of the effect of LNP charge after IM and IV administration.** To further understand the influence of LNP charge and protonation in the endosomal pH range, we tested

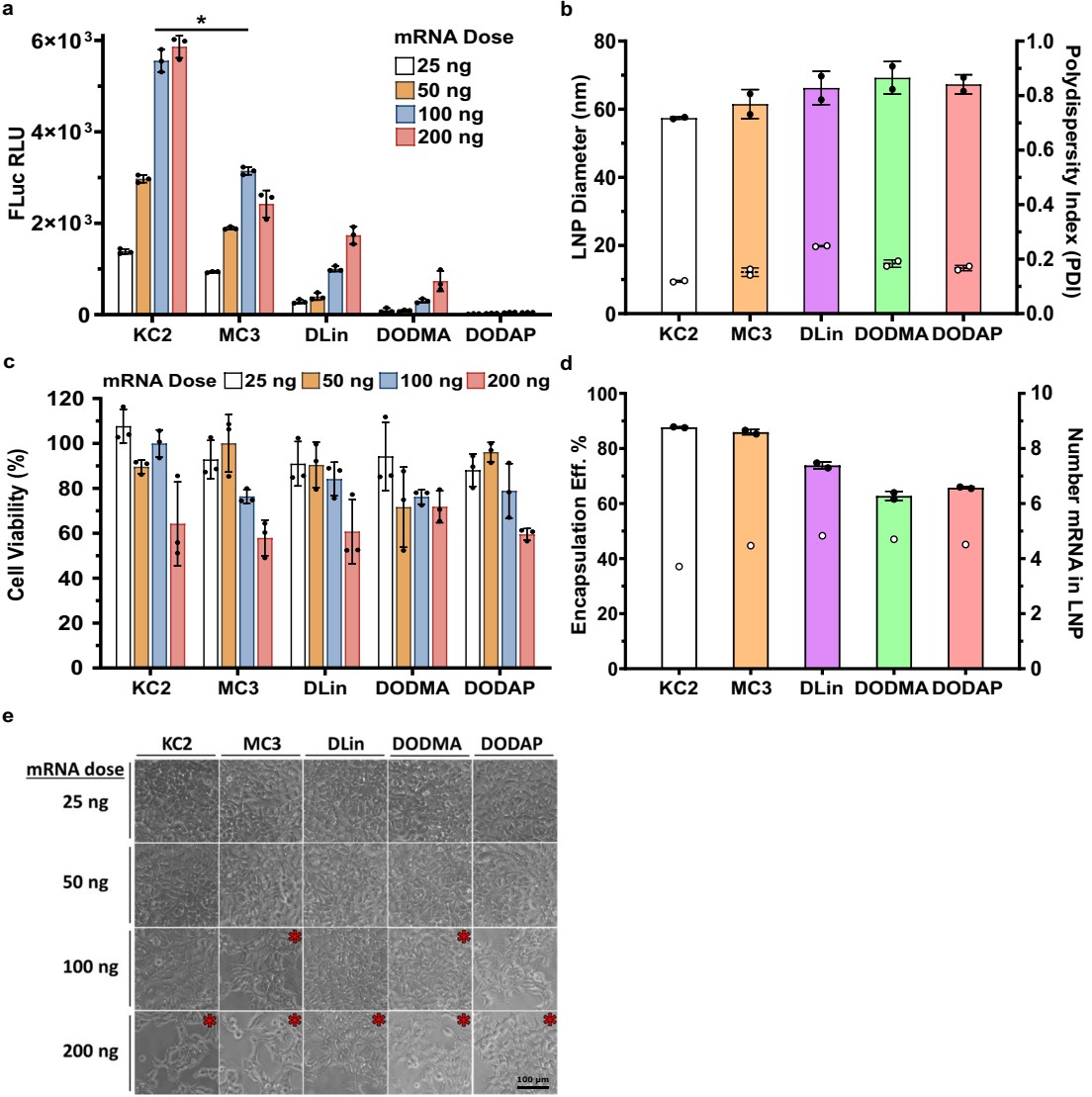

**Fig. 3 In vitro transfection efficiency, toxicity, size, and encapsulation efficiency of FLuc mRNA LNPs at NP lipid:mRNA ratio 4 prepared with five commercially available ionizable lipids DLin-KC2-DMA, DLin-MC3-DMA, DLin-DMA, DODMA, and DODAP. a** Transfection Efficiency by luciferase assay showed LNPs were able to deliver and express FLuc provided the TNS binding assay p$K$a was in the range 6.5–7, corresponding zeta potential p$K$a of 5.7–6.2 while the aqueous soluble and theoretical p$K$a's were in the range of 8.6–9.4. Theoretical design of effective ionizable lipids could be based on having theoretical p$K$a's that are 2–3 points higher than desired LNP p$K$a's, in order to examine a wide design space of potential candidates. *$p < 0.0001$ using a multivariate (dose and LNP predicting FLuc) analyses. All pairwise comparisons were significant; we only indicate KC2 vs MC3 for clarity **b** Number-weighted average diameter (bars) and PDI (circles) of LNPs by DLS. **c** Cell Viability after transfection with LNPs with Alamar Blue. **d** Encapsulation Efficiency (bars) using the ribogreen assay and calculated number of mRNA copies (circles) in the LNP using the molecular volume model described in Supplementary Information. **e** Phase contrast cell morphology after 24 h transfection period with LNPs at different doses was consistent with Alamar Blue in C. Doses of 100 and 200 ng showed areas of cell loss (*) in DODAP, DODMA, MC3, DLin, and KC2 at these higher concentrations. Scale bar is 100 μm. Mean ± SD ($N = 3$).

KC2 mRNA LNPs with NP lipid:mRNA ratios in the range of 2–8 in both IV and IM administration, and found that LNPs with low NP lipid:mRNA ratios were more potent both in IM (Fig. 7a, d) and IV (Fig. 7b, e). Luciferase expression in vivo correlated negatively with NP lipid:mRNA ratios (Fig. 7g) and positively with in vitro potency (Fig. 7h). Aligning with our observation above that the more negative MC3 LNPs expressed more strongly off-target in liver after IM administration than KC2 LNPs (Fig. 4e), here we found an even greater off-target expression for KC2 LNPs when the NP lipid:mRNA ratio was reduced to 2 and LNPs were highly negative with a pI of 5.5 (Fig. 7c, f). The ratio of

photon flux in liver/muscle at 20 h for NP2 KC2 LNPs was >100% while for an NP lipid:mRNA ratio of 4 it was ~50%. Taking together these in vivo results with those above using the different ionizable lipids, we consistently found that a more negatively charged LNP (MC3 vs KC2 and low NP lipid:mRNA vs high NP) will exhibit higher off-target expression in liver after IM administration (Figs. 4a, c and 7c, f). These results also highlight an important finding of our study that has impact on the rational design of LNPs, namely that charge-mediated targeting and trafficking of LNPs is influenced by the p$K$a of the ionizable lipid and by the ratio of charged components in the formulated LNP

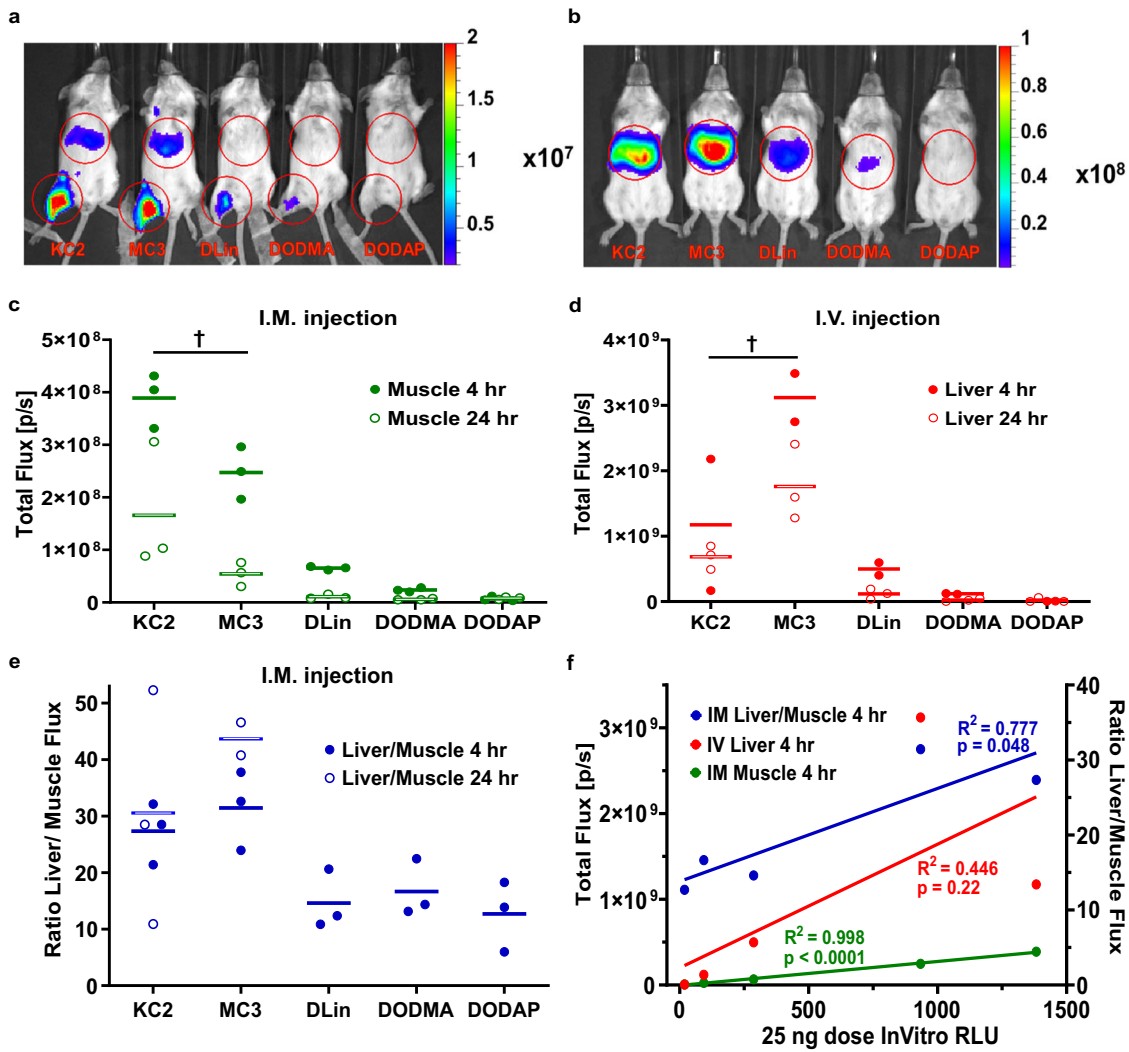

**Fig. 4 In Vivo Expression of FLuc mRNA in LNPS with ionizable lipids DLin-KC2-DMA, DLin-MC3-DMA, DLin-DMA, DODMA, and DODAP.** FLuc mRNA LNPs (5 μg) with NP lipid:mRNA ratio 4 were injected intramuscular (IM) or intravenous (IV) administration and imaged at 4 h and 24 h post-injection. **a** FLuc expression 4 h after IM administration and **b** 4 h after IV administration showing ROIs used to calculate Total Flux (photons/second). Scale bar is Radiance (p/sec/cm$^2$/sr). FLuc expression (Total Flux) for **c** IM administration (†$p < 0.055$) and **d** IV administration at 4 h and 24 h and **e** the ratio of Liver to Muscle photon flux for IM administration. KC2 was the most potent LNP for IM administration while MC3 with a pKa 0.4 points lower than KC2 was the most potent for IV administration. The ratio of liver to muscle expression after IM administration, an undesirable off-target effect, is highest for MC3 and is associated with a lower pKa, lower pI and a more negatively charged LNP than KC2. **f** In vitro FLuc expression was strongly predictive ($R^2 = 0.998$, $p < 0.0001$) of In Vivo FLuc IM expression at all doses shown here for 25 ng dose only, but not predictive of In Vivo FLuc IV expression ($R^2 = 0.446$, $p = 0.22$). Off-target expression in liver was also correlated to potency. Data and mean are shown in graphs. For IM administration of DLin, DODMA, and DODAP, expression in liver was too low for quantification.

that together determine the net absolute charge of the LNP as a function of pH.

## Conclusions

The direct measurement of the aqueous phase pKa of ionizable lipids by NMR permitted us to confirm the accuracy of theoretical pKa estimates and to develop a model that can relate these predicted pKa's to the mRNA-LNP pKa, which is one of the central parameters that control mRNA LNP delivery efficiency. We hypothesize these new theoretical and experimental tools will aid and accelerate the design of new ionizable lipids that have predictable ionization behavior in the LNP phase which in turn controls delivery efficiency and targeting. We also developed several new tools to analyze ionization behavior in the LNP including the ZP pKa that can provide net absolute

charge in the LNP through an electrokinetic model calculation as well as the LNP pI indicating the pH where LNPs are electrically neutral. When comparing in vitro potency of LNPs containing different ionizable lipids, we needed to consider multiple parameters to understand the potency, including LNP pKa, LNP ultrastructure, and the ionizable lipid structure as it relates to the molecular shape hypothesis. We found in vitro potency to strongly predict in vivo potency for intramuscular administration but not for intravascular administration, possibly due to charge-mediated binding of targeting mediators such as extracellular matrix in IM versus ApoE in IV administration. Off-target expression in liver after IM administration was also accentuated for more negatively charged LNPs in two different contexts: using distinct ionizable lipids with different pKa's and by adjusting the charge ratio of the ionizable lipid to mRNA. Additional findings that will aid future development

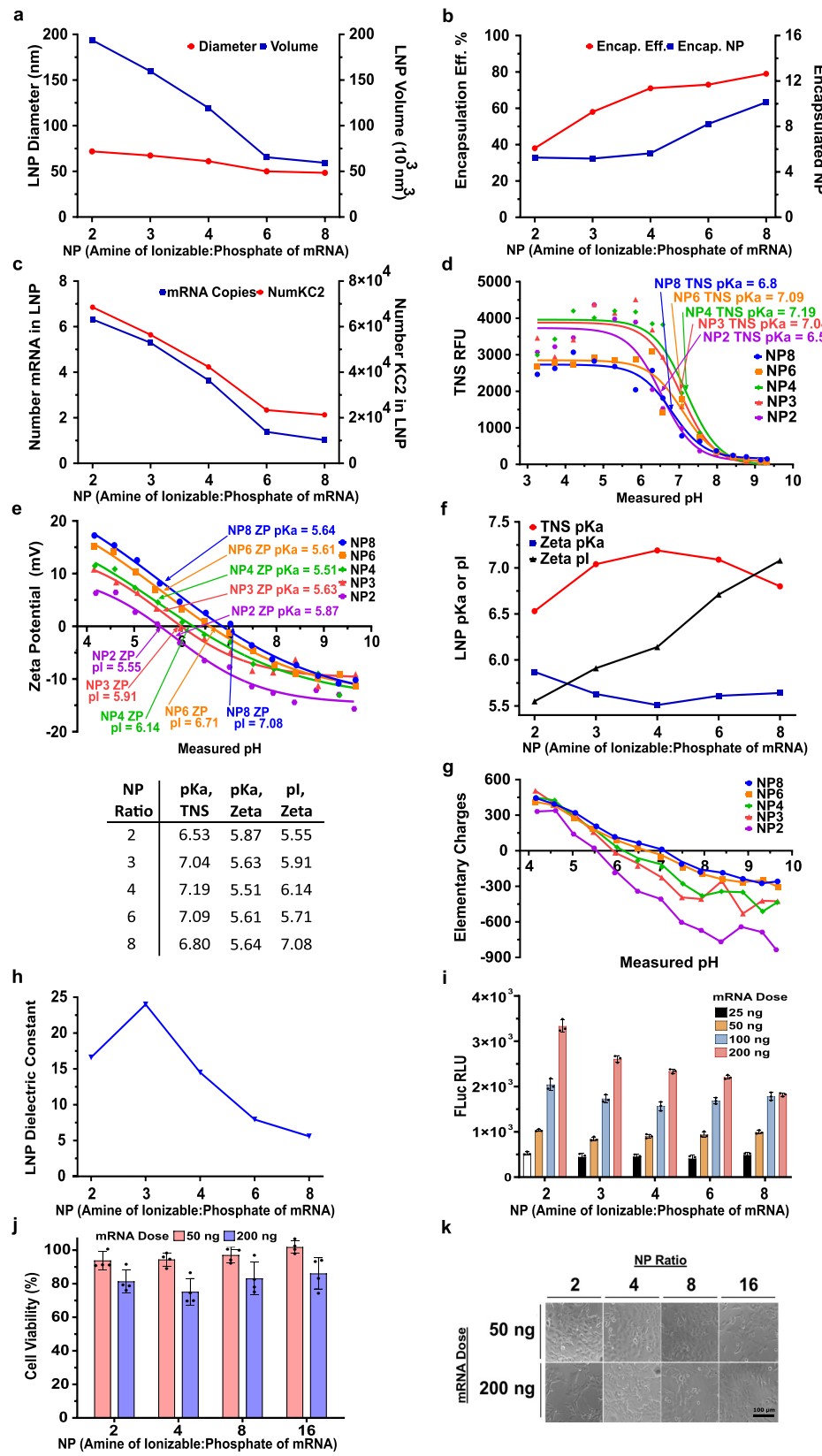

| NP Ratio | pKa, TNS | pKa, Zeta | pI, Zeta |
|---|---|---|---|
| 2 | 6.53 | 5.87 | 5.55 |
| 3 | 7.04 | 5.63 | 5.91 |
| 4 | 7.19 | 5.51 | 6.14 |
| 6 | 7.09 | 5.61 | 5.71 |
| 8 | 6.80 | 5.64 | 7.08 |

of mRNA-LNPs are the potential for TNS binding assay and ZP titrations to indicate endosomal protonation that correlates to potency, and the insight provided by a new model for estimating mRNA copy number and lipid numbers per LNP. These new tools and insights expand our ability to rationally design mRNA-LNPs that are potent and can be distributed to singly- or multi-targeted locations in vivo. A specific and important application of these new insights is in the reduction of systemic distribution and off-target expression after IM vaccine delivery.

**Fig. 5 DLin-KC2-DMA FLuc mRNA LNPs with NP lipid:mRNA ratio in the range 2–8 have NP lipid:mRNA ratio dependent size, mRNA copy number, encapsulation efficiency, pKa, and in vitro transfection efficiency.** FLuc mRNA KC2 LNPs assembled with NP lipid:mRNA ratio 2–8 (mole ratio of the ionizable lipid KC2 amine to the mRNA phosphate) were produced with lipid mole ratios 50:10:38.5:1.5 (KC2:DSPC:Cholesterol:PEG-DMG2000). **a** LNP diameter (number average DLS diameter) and volume decreased with increasing NP. **b** Encapsulation Efficiency increased at higher NP. Encapsulated NP lipid:mRNA ratio in the LNP (NP ÷ Encapsulation Efficiency) was double the formulation NP lipid:mRNA ratio at low N:P 2 (~5) and only ~25% higher (10 vs 8) at high NP lipid:mRNA ratio 8. **c** The average number of mRNA copies and of each of the 5 lipids in the LNP was calculated based on the molecular volume model described in Supplementary Information. The number of mRNA copies is proportional to volume of the LNP and inversely proportional to the Encapsulated NP lipid:mRNA ratio resulting in one copy per LNP at NP 8 and 6 copies at NP 2. **d** TNS binding assay pKa were found by fitting TNS fluorescence data to the HH equation (lines). The TNS binding assay only detects initial LNP protonation at higher pH > 6. **e** The LNP zeta potential (ZP) measured over the same pH range 4–10 was shifted upwards as NP increased. Unlike the TNS binding assay, ZP detects protonation continually down to pH 4 revealing ionizable lipid protonation in the LNP at pH values corresponding to intracellular endosomes and lysosomes. pKa from ZP was found by fitting to the extended HH model. pI was found by interpolating ZP to zero potential. **f** pKa from TNS and ZP did not change significantly with NP lipid:mRNA ratio however pI from ZP strongly increased with NP lipid:mRNA ratio by charging the LNP positively. **g** LNP net charge was calculated from ZP and the DLS number size using an electrokinetic model described in Supplementary Information. **h** An estimate of the LNP dielectric constant was obtained by calculating a theoretical LNP charge at high pH equal to one negative charge per nucleotide (see Supplementary Information) and dividing by the charge (**g**) measured at high pH determined from ZP. Resulting LNP dielectric constants ranged from 6 to 24, consistent with a lipid phase (dielectric constant 5[40]) containing some water[30] (dielectric constant 80). **i** Low NP LNPs exhibited a more linear dose response and were more potent at the higher doses than higher NP LNPs. **j** A slight reduction in cell viability measured with Alamar Blue was observed at the higher dose 200 ng but with no dependence on NP. **k** Cell morphology versus dose and NP was consistent with Alamar Blue in **j** showing some loss of viability at nigh 200 ng dose. Scale bar is 100 µm. Mean ± SD (N = 3) for **i**, **j**.

## Materials and methods

**Materials.** DLin-KC2-DMA was purchased from Biofine International INC. DLin-MC3-DMA was purchased from Advanced ChemBlock. DLin-DMA was purchased from Chem Scene. DMG-PEG (MW 2000) (DMG-PEG2000) was purchased from NOF America. Cholesterol was purchased from Combi Blocks. 1,2-dioleyloxy-3-dimethylaminopropane (DODMA), 1,2-dioleoyl-3-dimethylammonium-propane (18:1 DAP, DODAP), and 1,2-distearoyl-sn-glycero-3-phosphocholine (18:0 PC, DSPC) were purchased from Avanti Polar Lipids. Water-soluble analogs of DLin-KC2-DMA, DLin-MC3-DMA, DLin-DMA, DODMA, and DODAP were synthetized and purified as described below. Microfluidic cartridges compatible with the Spark NanoAssmblr™ were purchased from Precision Nanosystems. 3M Sodium Acetate pH = 5.5 was purchased from Thermofisher Scientific. The ONE-Glo Luciferase Assay System was purchased from Promega Corporation. Slide-A-Lyzer™ MINI Dialysis Device 0.5 ml (MWCO, 10 KDa), 20X TE Buffer, RNAse-free, and Quanti-iT Ribogreen RNA reagent were purchased from Thermo Fisher Scientific. Amicon Ultra-0.5 ml Centrifugal Filter Units, Triton X-100 and 6-(p-toluidino)-2-naphthalenesulfonic acid sodium salt (TNS) was purchased from Sigma. Codon optimized firefly luciferase (FLuc) sequence was cloned into an mRNA production plasmid (optimized 3′ and 5′ UTR and containing a 101 polyA tail), in vitro transcribed using N1-methylpseudouridine modified nucleoside, co-transcriptionally capped using the CleanCap technology (TriLink) and cellulose purified[42] to remove dsRNA. Purified mRNA was ethanol precipitated, washed, resuspended in nuclease-free water, and subjected to quality control (electrophoresis, dot blot, and transfection into human DCs). D-Luciferin (sodium salt) was purchased from REGIS technologies, INC. HEK293 cells were purchased from ATCC.

**Preparation of mRNA Lipid Nanoparticles.** mRNA-loaded LNPs were formulated using total lipid concentration of 50 mM comprised of commercial ionizable lipids (DLin-KC2-DMA, DLin-MC3-DMA, DLin-DMA, DODMA, and DODAP)/DSPC/Cholesterol/DMG-PEG2000 in the % mole ratios 50/10/38.5/1.5. Each lipid was dissolved in ethanol and mixed to reach the specified molar ratios in the organic phase. FLuc mRNA in the aqueous phase was dissolved in 43–50 mM sodium acetate buffer pH 4 to reach 1 mg/ml, keeping the NP lipid:mRNA ratio (moles amine of the ionizable lipid: moles phosphate of the mRNA) constant at 4, prior to mixing in the Spark NanoAssmblr™ (Precision NanoSystems). A 16-µL aliquot of the organic phase and a 32-µL aliquot of the aqueous phase were mixed and ejected into 48 µL of PBS at pH 7.4. The LNPs were then diluted into an additional 96 µL DPBS solution at pH 7.4 and dialyzed against PBS to reach pH 7.3–7.4 after 4–6 buffer exchanges over 4–6 hours using a Slide-A-Lyzer MINI Dialysis Device (MWCO, 10 KDa). To prepare formulations at different NP lipid:mRNA ratios (2–16) containing KC2/DSPC/Cholesterol/DMG-PEG2000 (%mole ratios 50/10/38.5/1.5), the lipids were dissolved in ethanol to reach a total lipid concentration of 6.25–100 mM and formulated with FLuc mRNA at 0.25–1 mg/mL in 21.5–50 mM sodium acetate buffer pH 4, as described above. After formulation, LNPs were then dialyzed against PBS pH 7.4 as above. For in vivo injections, LNPs were concentrated after dialysis using Amicon Ultra-0.5 mL Centrifugal Filter Unit.

**Ribogreen Assay for mRNA Encapsulation Efficiency.** Tris (10 mM, pH = 7.5)/EDTA (1 mM) (TE) and Triton/TE (2% v/v Triton in TE Buffer) were added in duplicates to a black microplate. Total mRNA in the LNP was diluted to ~4 ng/µL in TE and added to each TE and TE/Triton well in a 1:1 volume ratio. Two standard curves were included in the Ribogreen Assay, one containing mRNA and TE, and the other containing mRNA and Triton/TE. Each standard curve was used to calculate the mRNA concentration in each respective buffer. This approach using two standard curves is required for calculating the encapsulation efficiency and mRNA concentrations, in comparison to a single standard curve in Triton/TE, which can overestimate encapsulation by 5–10%, since Ribogreen has higher background fluorescence in Triton/TE vs TE. Microplates were incubated at 37 °C for 10 min to extract LNPs with Triton. Ribogreen reagent in DMSO was diluted 1:100 in TE Buffer and added to each well in a 1:1 volume ratio. Microplates were immediately introduced into the Cytation 5 Cell Imaging Multi-Mode Reader (Biotek) to read Fluorescence (Ex485/Em528).

**TNS binding assay.** LNP pKa was determined using the TNS binding assay. The TNS reagent was prepared as a 300 µM stock solution in DMSO. Following Zhang et al.[43], LNPs were diluted to 75 µM ionizable lipid, TNS to 6 µM in a total volume of ~93 µL of buffered solutions containing 20 mM boric Acid, 10 mM imidazole, 10 mM sodium acetate, 10 mM glycylglycine, 25 mM NaCl, and where the pH ranged from 3 to 10. The Cytation 5 Cell Imaging Multi-Mode Reader (Biotek) was used to read Fluorescence (Ex321/Em445). The pH was measured in each well after TNS addition. Mathematica (Wolfram Research) was used to fit the fluorescence data to the Henderson–Hasselbalch equation $RFU = RFU_{max} - (RFU_{max} - RFU_{min})/(1 + 10^{pKa-pH})$ to provide the pKa. We also tested two alternative TNS binding assay protocols (Supplementary Figs. 1 and 2 and Supplementary Tables 1 and 2) Sabnis et al.[44], where LNPs were diluted to 24 µM and TNS to 6.3 µM, and Jayaraman et al.[24], where LNPs were diluted to 40 µM and TNS to 1 µM.

**mRNA LNP size, zeta potential, ZP pKa, net charge, number of mRNA copies, and dielectric constant using dynamic light scattering and electrophoretic mobility.** LNPs were diluted to 6.25 ng/µL total mRNA in PBS pH = 7.4 and transferred into a quartz cuvette (ZEN2112) to measure size by Dynamic Light Scattering (DLS) in the Zetasizer Nano ZS (Malvern Panalytical) using particle RI of 1.45 and absorption of 0.001 in PBS at 25 °C with viscosity of 0.888 cP and RI of 1.335. Measurements were made using a 173° backscatter angle of detection previously equilibrated to 25 °C for 30 s in duplicates, each with 5 runs and 10 s run duration, without delay between measurements. Each measurement had a fixed position of 4.65 mm in the quartz cuvette with an automatic attenuation selection. Data were analyzed using a General-Purpose model with normal resolution. Diameter are reported as the number-average. The molecular volume model described in Supplementary Information was used to estimate the number of mRNA copies as well as the number of each of the 4 lipid components in the LNP using the number average diameter to calculate LNP volume. LNPs were diluted into the TNS buffer described above at pH ranging from 3 to 10 for ZP measurement by Electrophoretic Light Scattering (ELS) in the Zetasizer Nano ZP (Malvern Panalytical) using the same material and dispersant parameters described above, and the Smoluchowski model. Each measurement had voltage set manually at 80 Volts to avoid ohmic heating that occurred if voltage was set automatically. Measurements in the disposable folded capillary cuvette were made in triplicates for 20 runs each, and 30 s delay between each replicate. Mathematica was used to fit ZP data to the data to the extended[28] Henderson–Hasselbalch equation $\Psi = \Psi_{max} - (\Psi_{max} - \Psi_{min})/\left(1 + 10^{\frac{pKa-pH}{n}}\right)$ to provide the pKa, n and low pH and high pH ZP limits $\Psi_{max}$ and $\Psi_{min}$, respectively. The extended model is used here, since the LNP pKa is protonation dependent in a

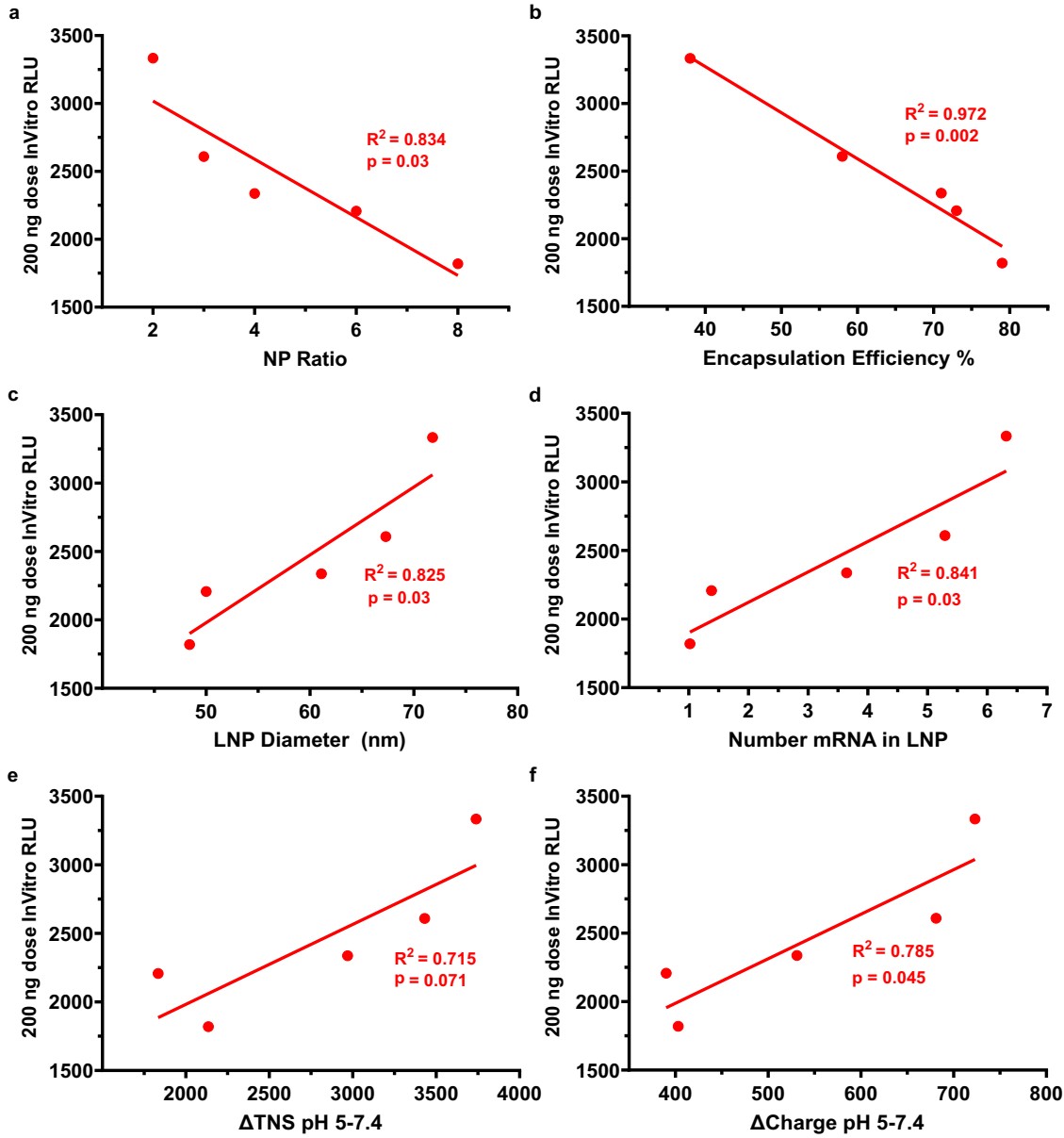

**Fig. 6 NP lipid:mRNA ratio dependent potency correlates with several structural and ionization characteristics of the LNPs.** High dose (200 ng total mRNA) potency of LNPs correlates negatively with **a** NP lipid:mRNA ratio, **b** Encapsulation Efficiency and positively with **c** Diameter, **d** mRNA copy number, **e** increase in TNS signal from pH 7.4 to 5 and **f** increase in zeta potential from pH 7.4 to 5. Reducing the NP lipid:mRNA ratio generates LNPs that are larger, contain more copies of mRNA per LNP (although a lower % of total mRNA) and have larger increases in both surface protonation (TNS in **e**) and total protonation (ZP in **f**) when the pH drops from physiological through the endosomal pH range to pH 5. These lower NP LNPs are richer in mRNA vs lipid and have greater protonation in the endosome that may be responsible for increasing their potency.

way similar to a polyelectrolyte[28,29]. TNS binding assay data did not require the extended model since TNS dye binding only detects LNP surface charge. Equations 20–24 of the model described in supplementary information were used to calculate net LNP charge from electrophoretic mobility at each pH using the full non-linear form of the Poisson–Boltzmann equation to relate surface potential to charge and a Henry's function, without the Smoluchowski approximation. The calculated charge was then fit to the extended Henderson–Hasselbalch equation to estimate the high pH charge limit, which then permitted the LNP dielectric constant to be estimated according to Eq. 26 in Supplementary Information. pH-dependent ion (hydroxide and hydronium) binding to the LNP was apparent when measuring ZPs of charge neutral LNPs and was not considered when calculating LNP charge described above (Supplementary Fig. 4).

**Cryoelectron microscopy and small-angle x-ray scattering**. Grids for electron microscopy were plunge-frozen using a Vitrobot Mark IV system. The chamber was set to 25 °C and 100% humidity. LNPs (3 μL) were applied to grids, incubated for 1 minute, and blotted for 5.5 s before plunging into liquid ethane. No drain time (post-blotting) was used prior to plunging. The lowest blot force that produced consistent grids was used (10–15 on our system). Grids were imaged on a Talos Arctica system (Thermo Fisher Scientific) at 200 kV with a Falcon 3EC detector, using EPU for data collection. The nominal magnification was 45,000x, with a calibrated pixel size of 0.223 nm. Images were collected in integrating mode using 5s exposures, with a total dose of 60 e/Å², and 66 fractions which were motion-corrected using Motioncor2[45].

Solution X-ray scattering data were acquired using the SAXSLAB Ganesha instrument using incident radiation with the wavelength $\lambda = 1.542$ Å produced by the Rigaku MicroMax 007HF rotating anode generator. Dectris Pilatus 300 K area detector was employed to register the scattered radiation with the transmitted intensity monitored via the pin diode. Samples were kept at 25 °C and exposed for the duration of 10 sequential 900-s frames. Pixel intensity outliers due to the background radiation were removed and the data corrected for the detector sensitivity profile and the solid angle projection for each pixel. The data were

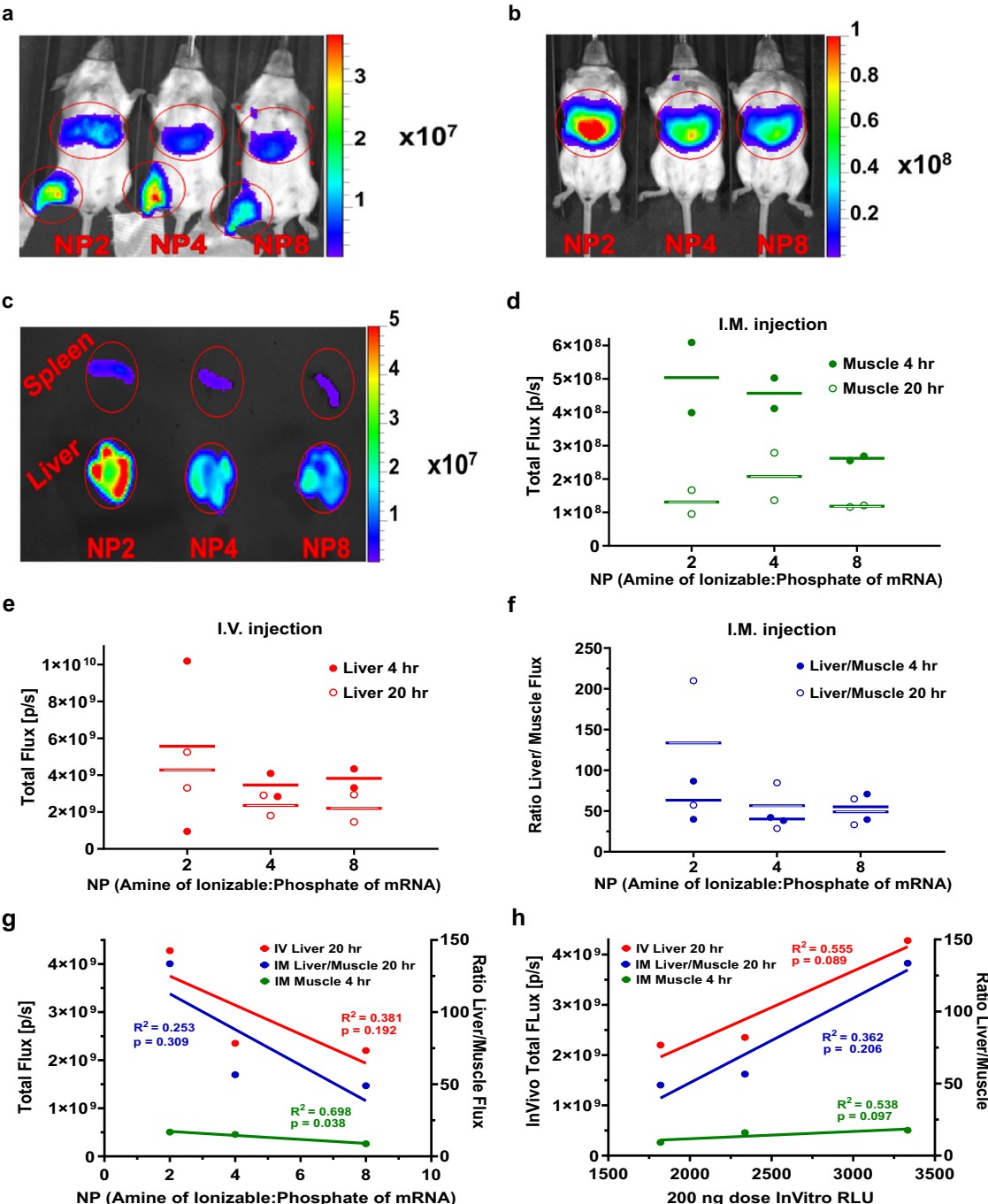

**Fig. 7 In vivo expression of FLuc mRNA (5 μg total mRNA) in KC2 LNPs with NP lipid:mRNA ratio in the range 2–8 for intramuscular (IM) and intravenous (IV) administration at 4 h and 20 h post-injection. a** FLuc expression 20 h after IM administration and **b** 4 h after IV administration and **c** ex vivo liver and spleen from IV at 20 h. Color scale bars are Radiance (p/sec/cm2/sr). Total flux (photons/second) quantified from ROIs at 4 h and 20 h for **d** IM administration, **e** IV administration, and **f** the photon flux ratio liver:muscle for IM injections. **g** IM 4 h, Liver/Muscle IM 20 h, IV 20 h correlate negatively with NP. **h** IM 4 h, Liver/Muscle IM 20 h, IV 20 h correlate positively with in vitro potency at 200 ng. Efficiency of expression was higher for lower NP lipid:mRNA ratio in both IV and IM administration as well as in vitro at 200 ng dose. Off-target expression in the liver after IM administration was found at 4 h and 20 h and was >100% that in muscle for NP lipid:mRNA ratio 2 showing rapid systemic dissemination and expression of these low NP negatively charged KC2 LNPs upon IM administration. Data and mean are shown in graphs.

converted to one-dimensional scattering intensity curves, frame-averaged, and buffer-subtracted. Data sets acquired at sample-detector geometries of 1755 mm and 435 mm, and slit openings of 0.2 mm and 0.8 mm, respectively, were merged to extend the accessible $q$-range from 0.0055 Å$^{-1}$ to 0.8 Å$^{-1}$, where $q = 4\pi\sin(\theta)/\lambda$.

**In vitro transfection, luciferase assay, and cell viability assay**. HEK293 cells were seeded in white 96-well plates at a density of $12 \times 10^3$ cells per well in 100 μL

EMEM medium (10% FBS) the day before transfection and incubated at 37 °C 5% $CO_2$. FLuc mRNA-loaded LNPs were diluted so that 8, 16, 24, and 32 μL volumes contained 25, 50, 100, and 200 ng mRNA FLuc and HEK293 cells were transfected using triplicates for each of the 4 doses 24 h after seeding. After a further 24 h for transfection, 100 μL of One-Glo substrate was added directly to the wells to detect luciferase expression based on luminescence in Cytation 5 luminometer plate reader. A toxicity assay was performed 24 h after transfection in a separate plate, where transfected cells were incubated with pre-warmed Presto Blue HS reagent

(10% v/v) for 15 min at 37 °C. Microplates were immediately introduced into the Cytation 5 to read Fluorescence (Ex540/Em590).

**In vivo live animal imaging for luciferase expression following intramuscular, intravenous, and intradermal administration of FLuc mRNA LNPs.** The investigators faithfully adhered to the "Guide for the Care and Use of Laboratory Animals" by the Committee on Care of Laboratory Animal Resources Commission on Life Sciences, National Research Council. Mouse studies were conducted under protocols approved by the Institutional Animal Care and Use Committees (IACUC) of the University of Pennsylvania (UPenn). All animals were housed and cared for according to local, state, and federal policies in an Association for Assessment and Accreditation of Laboratory Animal Care International (AAA-LAC)-accredited facility. 8-week old female Balb/c mice purchased from Charles Rivers and acclimatized at the University of Pennsylvania for 7 days before experiments. Mice were injected with 5 µg FLuc mRNA-LNP using a 3/10 cm³ 29 G insulin syringe (BD Biosciences). mRNA-LNPs were diluted in PBS and injected into the gastrocnemius muscle (50 µL injection volume) or in the retroorbital sinus for intravenous (IV) administration. At 4 h post-injection, mice were anesthetized with 2% isofluorane in oxygen and imaged 10 min after intraperitoneal injection of 250 µL D-Luciferin (15 mg/ml). Bioluminescence imaging was performed using an IVIS Spectrum imaging system (Caliper Life Sciences). Imaging was repeated at 24 h, animals euthanized, and organs collected for ex vivo imaging.

**Synthesis of water-soluble ionizable lipid analogs.** Solvents were purchased from Sigma Aldrich, Combi blocks, Oakwood chemicals, Alfa Aesar, VWR, and Thermofisher. Anhydrous methylene chloride (DCM), anhydrous tetrahydrofuran (THF) anhydrous DMF were purchased from Sigma Aldrich. Product purification was done using SiliaFlash® Irregular Silica Gel, F60 40–63 µm, 60 Å and SiliaPlate™ thin layer chromatography Plates (TLC), Glass-Backed Silica, Opt. KMnO₄, 250 µm, 20 × 20 cm, and F254 plates purchased from Silicycle. NMR spectra were recorded on a Bruker 400 MHz Spectrometers using CDCl₃, D₂O (Sigma Aldrich), as d-solvents and internal standards (δ 7.26 for ¹H NMR and δ 77.00 for ¹³C NMR). Solution of 1 M NaOH and 1 M HCl were purchased from Sigma Aldrich. NMR Spectra for synthesized compounds are in Supplementary Fig. 5.

a) DODAP water-soluble analog

DODAP water-soluble analog (**3** above) was synthesized as per the previous reports[46,47]. 3-(dimethylamino)propane-1,2-diol (**1a**) (16.8 mmol, 2.0 g, 1.99 mL), triethylamine (42 mmol, 4.25 g, 5.83 mL), and CH₂Cl₂ (200 mL) were placed in a 250-mL two-neck round bottom flask and cooled to 0 °C in an ice bath. The reaction was stirred for 1 h and then pivaloyl chloride (**2a**) (37.8 mmol, 4.55 g, 4.65 mL) was added dropwise and the reaction stirred for 7 h. The reaction progress was monitored every 3 h by TLC (chloroform/methanol 9:1 v/v, can be visualized with iodine stain). After complete consumption of 1a, the reaction was quenched by adding water. CH₂Cl₂ was concentrated by rotary evaporation. The mixture was dissolved in 200 mL of CH₂Cl₂ and washed with 150 mL of water and 150 mL of saturated NaHCO₃ solution. The organic phase was dried over magnesium sulfate and evaporated. The crude product was purified on a silica gel column eluted with chloroform containing 0–2% methanol. Column fractions were analyzed by thin layer chromatography (TLC) and fractions containing pure product ($R_f = 0.5$) were pooled and concentrated, to obtain the product as yellow oil (**3**) (3.56 g, 62% yield). HNMR data (400 MHz, CDCl₃, δ = 7.26 ppm as standard): δ 5.17-5.15 (m, 1H), 4.38-4.34 (m, 1H), 4.09 (dd, 6.44 Hz, 6.44 Hz, 1H), 2.45 (dddd, 6.68 Hz, 6.68 Hz, 5.92 Hz, 5.80 Hz, 2H), 2.26 (s, 6H), 1.18 (s, 18H). ¹³C NMR: (100 MHz, CDCl₃, δ = 77.00 ppm as standard): δ 178.1 (C), 177.7 (C), 69.4 (CH), 64.0 (CH₂), 59.2 (CH₂), 46.0 (2 × CH₃), 38.8 (C), 38.7 (C), 27.1 (6 × CH₃).

b) DLin-DMA/DODMA water-soluble analog

DLinDMA/DODMA water-soluble analog (**5** above) was synthesized as per the previous reports[26,48–50]. Excess sodium hydride NaH (8.3 equiv, 140 mmol, 3.34 g, 60% in oil) was added into a 250-mL two-neck round bottom (RB) flask under nitrogen atmosphere and anhydrous DMF (10 mL) was added. The resulting suspension was stirred for 10 min. To this slurry, a solution of 3-(dimethylamino)-propane-1,2-diol (**1a**) (1.0 equiv, 16.8 mmol, 2.0 g) in DMF (15 mL) was added dropwise for 10 min at 0 °C. The resulting suspension was heated to reflux for 24–36 h. Pentyl bromide (**4a**) (2.50 equiv, 42.0 mmol, 5.2 ml) in DMF (10 mL) solution was prepared and added to the reaction mixture at 0 °C. After addition, the

reaction was heated to reflux for 5 days and then cooled to room temperature and the mixture filtered through a 3-cm plug of celite and washed with methylene chloride (200 mL). The filtrate was removed under vacuum and the residue was partitioned between methylene chloride (100 mL), and brine (50 mL), used to aid phase separation. The two layers were separated and the organic layer was washed with dried over anhydrous magnesium sulfate. The crude product was purified on a silica gel column eluted with chloroform containing 0–5% methanol. Column fractions were analyzed by thin layer chromatography (TLC) (silica gel, chloroform/methanol 9:1 v/v, can be visualized with iodine or PMB stain) and fractions containing pure product ($R_f = 0.5$) were pooled and concentrated, to obtain product as brown oil (**5**) (1.92 g, 37% yield). ¹H-NMR: (400 MHz, CDCl₃, δ = 7.26 ppm as standard): δ 3.58-3.54 (m, 1H), 3.52-3.47 (m, 2H), 3.44-3.40 (m, 4H), 2.43-2.33 (m, 2H), 2.25 (s, 6H, 2×NCH₃), 1.55 (t, 6.34 Hz, 4H), 1.31-1.29 (m, 8H), 0.89-0.86 (m, 6H), 0.86 (t, 6H, 2 × CH₃). ¹³C NMR: (100 MHz, CDCl₃, δ = 77.0 ppm as standard): δ 77.2 (C), 72.1 (CH₂), 71.5 (CH₂), 70.2 (CH₂), 61.1 (CH₂), 46.3 (2 × CH₃), 29.8 (CH₂), 29.3 (CH₂), 28.3 (2 × CH₂), 22.5 (2 × CH₂), and 14.0 (2 × CH₃).

c) DLin-MC3-DMA water-soluble analog

DLin-MC3-DMA water-soluble analog (**8** above) was synthesized as per the previous literature reports[51,52]. A mixture of 4-(dimethylamino)butanoic acid hydrochloride salt (**6a**) (59.7 mmol, 1.0 g), 1-(3-dimethylaminopropyl)-3-ethylcarbodiimide hydrochloride (EDCi•HCl) (7.16 mmol, 0.63 g), 4-(dimethylamino)pyridine (DMAP) (1.49 mmol, 0.18 g), triethylamine (42.0 mmol, 4.25 g, 5.83 mL), and CH₂Cl₂ (100 mL) were placed in a 250-mL two-neck round bottom flask. The reaction was stirred for 20 minutes at room temperature and neopentyl alcohol (**7a**) (7.16 mmol, 0.78 mL) was added dropwise and the reaction stirred overnight. Reaction progress was monitored by TLC (chloroform/methanol 9:1 v/v, can be visualized with iodine stain), after complete consumption of **6a** the solvent CH₂Cl₂ was removed by rotary evaporation. The mixture was dissolved in 100 mL of CH₂Cl₂ and washed with 150 mL of water and 150 mL of saturated NaHCO₃ solution. The organic phase was dried over magnesium sulphate and evaporated. The crude product was purified on a silica gel column eluted with chloroform containing 0–1% methanol. Column fractions were analyzed by thin layer chromatography (TLC) and fractions containing pure product ($R_f = 0.4$) were concentrated, to obtain product as yellow oil (**8**). (0.58 g, 48% yield). ¹H-NMR: (400 MHz, CDCl₃, δ = 7.26 ppm as standard): δ 2.36 (t, 7.4 Hz, 2H), 2.31 (t, 7.2 Hz, 2H), 2.22 (s, 6H, 2×NCH₃), 1.84-1.78 (m, 2H), 0.86 (s, 9H, 2 × CH₃). ¹³C NMR: (100 MHz, CDCl₃, δ = 77.0 ppm as standard): δ 173.6 (C), 73.6 (CH₂), 58.8 (CH₂), 45.3 (2 × CH₃), 32.1 (CH₂), 31.2 (C), 26.4 (9 × CH₃), and 22.9 (2 × CH₂).

d) DLin-KC2-DMA water-soluble analog

*Step-1*: DLin-KC2-DMA water-soluble analog (**11** above) was synthesized as per previous literature reports[53,54]. To a solution of 2-(2,2-dimethyl-1,3-dioxolan-4-yl) ethanol (**9a**) (800 mg, 5.47 mmol) in methylene chloride (25 mL) in a 1000-mL RBF at cooled to 0 °C was added triethylamine (0.915 mL, 6.57 mmol), 4-(dimethylamino)pyridine (DMAP) (134 mg, 1.10 mmol), and tosyl chloride (1.10 g, 5.75 mmol). The reaction was stirred at room temperature overnight. The reaction was dried over Na₂SO₄ and concentrated. Ethyl ether 50 ml was added to the crude product and the filtrate was concentrated to obtain colorless oil. The compound was used directly to the next step without purification (**10a**) (1.06 g, 65%).
*Step-2*: a Biotage microwave vial was equipped with a stir bar, 2-(2,2-dimethyl-1,3-dioxolan-4-yl)ethyl 4-methylbenzenesulfonate (**10a**) (500 mg, 1.66 mmol), and 1.0 M diethylamine in methanol (4 mL). The vial was purged with N₂ gas, seal capped, and heated at 120 °C for 10 min in a Biotage Initiator microwave. The reaction was diluted with ethyl acetate and water. The layers were separated and the aqueous layer was extracted (30 mL) with ethyl acetate. The combined organics were dried over Na₂SO₄, filtered and concentrated by rotary evaporation. The residue was of sufficient purity for determination of pKa by NMR without further purification. (**11**) (0.21 g, 75%).¹H-NMR: (400 MHz, CDCl₃, δ = 7.26 ppm as standard): δ 4.15-4.01 (m, 2H), 3.52 (t, 7.2 Hz, 2H), 2.41-2.23 (m, 2H), 2.20 (s, 6H, 2×NCH₃), 1.81-1.62 (m, 2H), 1.38 (s, 3H, CH₃), 1.32 (s, 3H, CH₃). ¹³C NMR: (100 MHz, CDCl₃, δ = 77.0 ppm as standard): δ 108.6 (C), 74.6 (CH), 69.4 (CH₂), 56.2 (CH₂), 45.5 (2 × CH₃), 31.9 (CH₂), 26.9 (CH₃), and 25.7 (CH₃).

**NMR measurement of pKa of water-soluble ionizable lipid analogs.** The pH dependence of proton NMR chemical shifts was used to measure the pKa's of the ionizable lipid water-soluble analogs following published methods[27,55]. Chemical

shifts of piperazine and imidazole were used as internal pH indicators. Solutions were prepared with 100 mM KCl, 2 mM piperazine, 2 mM imidazole, and 5 mM water-soluble ionizable lipid analog in 95% $H_2O$-5% $D_2O$. This solution was split into two equal volumes and one titrated to a lower pH (e.g. 6) using 0.1 M HCl and the other to an upper pH (e.g. 12) using 0.1 M NaOH. Intermediate pH values were obtained by mixing different proportions of these two solutions. NMR measurements were performed on a Bruker 400 MHz spectrometer where $^1H$ spectra were acquired at each of ~20 pH values ranging from the lower to upper pH (e.g. 6–12). Chemical shifts from piperazine and imidazole were then used to calculate the pH of each solution according to published methods[27,55] and the chemical shifts of the N-terminal amine group protons of the water-soluble ionizable lipid analogs were fit to the Henderson–Hasselbalch equation $\delta = \delta_{max} - (\delta_{max} - \delta_{min})/(1 + 10^{pKa-pH})$ to provide the pKa of the N,N-Dimethylamine moiety in the different head groups (Spectra in Supplementary Fig. 1).

**Theoretical calculation of pKa.** Experimentally determined pKa values from NMR, ZP, and TNS binding assays were compared against theoretically calculated values using Advanced Chemistry Development, Inc. ACD/Labs software. The ACD/pKa database algorithmically estimates pKa values of whole molecules in an aqueous environment based on their constituent fragments using two approaches. The Classic algorithm employs a database of Hammett-type equations parameterized to cover most ionizable functional groups, each characterized by several equations involving variations of sigma constants. The Galas algorithm estimates pKa microconstants for all possible ionization centers in a hypothetical uncharged state based on the surroundings of the reaction center and neighboring ionization centers to produce microconstants from which pKa macroconstants are obtained. Classic algorithm calculations were used in this study.

**Statistics and reproducibility.** The Linear Least Squares Multivariate Model in the JMP Pro 15.1.0 software was used to perform comparisons between groups. For In Vivo IVIS imaging data the 4 and 20/24 h time points were considered repeated measures. Pearson's correlation coefficients were also calculated in JMP.

**Disclaimer.** Certain commercial equipment, instruments, materials, suppliers, or software are identified in this paper to foster understanding. Such identification does not imply recommendation or endorsement by the National Institute of Standards and Technology (NIST), nor does it imply that the materials or equipment identified are necessarily the best available for the purpose.

**Reporting summary.** Further information on research design is available in the Nature Research Reporting Summary linked to this article.

## Data availability

The source data for the graphs and charts in the figures is available as Supplementary data Files. The data that support the plots within this paper and other findings of this study are available from the corresponding author upon reasonable request.

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

## Acknowledgements

M.D.B. acknowledges financial support from the Commonwealth Research Commercialization Fund (CRCF) Center for Innovative Technology (CIT) Award ER17-002-LS.

## Author contributions

M.J.C., S.A., M.G.A., M.P., D.W., T.E.C., A.G. and M.D.B. conceived and designed the experiments and wrote the manuscript. M.J.C., S.A., T.E.C., A.G., H.S., L.W.S. and O.S performed experiments. All authors discussed the results and commented on the manuscript. M.D.B. directed the research.

## Competing interests

The authors declare no competing interests.
