## [Peer Review File · Communications Biology]

Reviewers' comments:

Reviewer #1 (Remarks to the Author):

Review Summary

Overall, the manuscript provides a detailed report on the generation of a novel approach to predict the pKa of lipid nanoparticles which influences their ability to be used as mRNA delivery devices in vitro and in vivo. The paper covers a topic of significant interest to the readership of Communications Biology and the scientific community more generally due to lipid nanoparticles being leveraged for some of the COVID-19 vaccines currently available. The paper is quite well written and only requires minor to moderate alterations before it is ready for publication. Please find details regarding these requested changes below.

Comments

1) Page 3 – The pH of late-stage endosomes especially after fusion with lysosomes is below 5 (i.e., ~ 4.5 - 5.0). If the authors are wishing to indicate the pH of endosomes pre-lysosomal fusion which are more in the 5.0 - 6.5 pH range, then please provide a bit more context to make this clear. Alternatively, the authors should state the lower pH which will be expected.

2) Page 4 – Please alter the text reading “LNPs targeted the lung” to either “LNPs that targeted the lung” or “LNPs targeting the lung”.

3) Page 4 – Please use “TNS binding assay” instead of “TNS” and make this change for all future mentions within the text.

4) Pages 8 - 9 – It is unclear how good of a job employing HEK293 cells derived from human embryonic kidney cells in vitro do in mimicking the relevant cell populations in vivo. For example, using primary or stem-cell derived hepatocytes instead of HEK293 as an analog for the intravenous delivery approach could provide helpful supplemental data. That being said, providing text mentioning the difference in cell populations, the option to enhance testing of the in vitro versus in vivo correlation in the future, and slightly softening the Apo-E absorption section since it was not explicitly tested would be sufficient for publishing this work. For the latter part, just making it clear that this is a possible explanation instead of the known explanation is acceptable.

5) Page 8 – It appears that the in-text citation for Figure 3, is labeled as Figure 4, meaning that Figure 3 is currently not cited in the text. Please fix this issue and double check to make sure all in-text figure mentions correspond with the correct figure and panel.

6) Page 10 - 11 – The authors clearly establish the different NP ratios are relating ionizable lipid amine to mRNA phosphate in the first sentence, but this is just described as “NP ratios” later in the text which could be confusing to the reader. The authors are suggested changing this to something like “NP lipid:mRNA ratios”, so it is clearer each time it is used in the text.

7) Page 10 – Please find another place to cite the lipid LNP dielectric constant as it now looks like the value should be 5 raised to the 40th power. Possibly moving this to after the word lipid or the parentheses would help.

8) Page 10 – Please modify “ratios and therefore higher mRNA content” to read “ratio, and therefore higher mRNA content,”.

9) Page 10 – Please change “(Fig 5 I)” to read “(Fig 5I)”.

10) Page 11 – Please modify the section header to read “IM administration and IV administration” instead of just “IM administration” since both delivery routes were utilized in this section.

11) Page 35, Figure 4, Panels C - F – It would be helpful to the reader if the color coding and dot

shapes followed a specific pattern. Maybe 4 hour and 24 hour of the same tissue analyzed could be in the same color pattern but different shading. The combined tissue could then be mixed coloring (e.g., blue pallet for muscle, red for liver, and purple for muscle/liver). Figure 4F, could then follow the same color scheme and shading as is seen in Figures 4C - 4E to help guide the readers eyes to the connections between these panels. Figure 4E is also a bit hard to understand what it is reporting. Is it the % Total Flux found in the liver as compared to the total of the values of the liver and the muscle? If so, maybe change the figure legend to better represent this. Also, please align the x-axes horizontally and y-axes vertically, so the figure panels orient better like was done with Figures 1 & 3. Finally, it is unclear to the reader why there were no Liver/Muscle data points for DLin, DODMA, and DODAP.

12) Page 39, Figure 7, Panels D - H – Please take a similar approach for color coding and dot shapes in these plots as you do with the updated Figure 4.

Reviewer #2 (Remarks to the Author):

In their manuscript, Carrasco et al conduct a thorough characterization of various LNP compositions, including chemical, biophysical, and biological properties in order to provide some insight into LNP properties that may explain differences in levels of mRNA-mediated gene expression as well as localization of mRNA delivery following intramuscular or intravenous delivery routes.

The chemical and biophysical characterization is very well done and provides a valuable data set of interest to those in the field. I only have one major criticism with regard to the biological characterization, particularly the comparisons in gene expression levels, both in vitro and in vivo. The overall differences in expression between groups appears quite small, although presenting the data on a log scale or providing fold-change data may help the readers discern the differences more accurately. Furthermore, no statistical analyses were performed and the number of biological replicates and experimental replicates were small so at least conclusions regarding in vivo delivery efficiency cannot be drawn with any degree of certainty.

Response to Reviewers

We thank the reviewers for their careful reading and analyses of our manuscript. We have taken their comments into consideration, responded point by point below, and revised the manuscript accordingly.

Reviewer #1 (Remarks to the Author):

Review Summary

Overall, the manuscript provides a detailed report on the generation of a novel approach to predict the pKa of lipid nanoparticles which influences their ability to be used as mRNA delivery devices in vitro and in vivo. The paper covers a topic of significant interest to the readership of Communications Biology and the scientific community more generally due to lipid nanoparticles being leveraged for some of the COVID-19 vaccines currently available. The paper is quite well written and only requires minor to moderate alterations before it is ready for publication. Please find details regarding these requested changes below.

Comments

1) Page 3 – The pH of late-stage endosomes especially after fusion with lysosomes is below 5 (i.e., ~ 4.5 - 5.0). If the authors are wishing to indicate the pH of endosomes pre-lysosomal fusion which are more in the 5.0 - 6.5 pH range, then please provide a bit more context to make this clear. Alternatively, the authors should state the lower pH which will be expected.

We have changed on line 63 :

“However, once inside the cell in endosomes the pH declines to near 5”

to :

“However, once inside the cell in endosomes the pH declines to near 4.5 prior to lysosomal fusion”

2) Page 4 – Please alter the text reading “LNPs targeted the lung” to either “LNPs that targeted the lung” or “LNPs targeting the lung”.

We have changed in line 76 to “LNPs targeting the lung”

3) Page 4 – Please use “TNS binding assay” instead of “TNS” and make this change for all future mentions within the text.

“TNS” was changed to “TNS binding assay” where appropriate throughout the manuscript.

4) Pages 8 - 9 – It is unclear how good of a job employing HEK293 cells derived from human embryonic kidney cells in vitro do in mimicking the relevant cell populations in vivo. For example, using primary or stem-cell derived hepatocytes instead of HEK293 as an analog for the intravenous delivery approach could provide helpful supplemental data. That being said, providing text mentioning the difference in cell populations, the option to enhance testing of the in vitro versus in vivo correlation in the future, and slightly softening the Apo-E absorption section since it was not explicitly tested would be sufficient for publishing this work. For the latter part, just making it clear that this is a possible explanation instead of the known explanation is acceptable.

We added the following on line 222:

HEK293 cells were chosen as a first model cell type for potency screening that could be replaced in future studies by primary cells more representative of in vivo targets.

We also inserted “possibly” into the following ApoE statement on line 84.

These particular LNPs apparently enter the vasculature after IM injection and subsequently express in liver hepatocytes possibly due to passive ApoE-mediated targeting²⁰

5) Page 8 – It appears that the in-text citation for Figure 3, is labeled as Figure 4, meaning that Figure 3 is currently not cited in the text. Please fix this issue and double check to make sure all in-text figure mentions correspond with the correct figure and panel.

We have corrected Figure 4 to Figure 3 on line 226.

6) Page 10 - 11 – The authors clearly establish the different NP ratios are relating ionizable lipid amine to mRNA phosphate in the first sentence, but this is just described as “NP ratios” later in the text which could be confusing to the reader. The authors are suggested changing this to something like “NP lipid:mRNA ratios”, so it is clearer each time it is used in the text.

We changed “NP” and “NP ratio” to “NP lipid:mRNA ratio” throughout the manuscript.

7) Page 10 – Please find another place to cite the lipid LNP dielectric constant as it now looks like the value should be 5 raised to the 40th power. Possibly moving this to after the word lipid or the parentheses would help.

The 40 in superscript was a citation reference. To remove this ambiguity we changed in line 284 :

The LNP dielectric constant ranged from 6-24, which is intermediate between that of lipid (5⁴⁰) and water (80)

to

The LNP dielectric constant ranged from 6-24, which is intermediate between that of lipid⁴⁰ (5) and water (80)

8) Page 10 – Please modify “ratios and therefore higher mRNA content” to read “ratio, and therefore higher mRNA content,”.

This change has been made on line 288.

9) Page 10 – Please change “(Fig 5 I)” to read “(Fig 5I)”.

This change has been made on line 291.

10) Page 11 – Please modify the section header to read “IM administration and IV administration” instead of just “IM administration” since both delivery routes were utilized in this section.

This change has been made on line 305.

11) Page 35, Figure 4, Panels C - F – It would be helpful to the reader if the color coding and dot shapes followed a specific pattern. Maybe 4 hour and 24 hour of the same tissue analyzed could be in the same color pattern but different shading. The combined tissue could then be mixed coloring (e.g., blue pallet for muscle, red for liver, and purple for muscle/liver). Figure 4F, could then follow the same color scheme and shading as is seen in Figures 4C - 4E to help guide the readers eyes to the connections between these panels. Figure 4E is also a bit hard to understand what it is reporting. Is it the % Total Flux found in the liver as compared to the total of the values of the liver and the muscle? If so, maybe change the figure legend to better represent this. Also, please align the x-axes horizontally and y-axes vertically, so the figure panels orient better like was done with Figures 1 & 3. Finally, it is unclear to the reader why there were no Liver/Muscle data points for DLin, DODMA, and DODAP.

We made color changes consistent with the reviewer’s suggestions making muscle data green, liver red and the ration of liver to muscle blue. Closed symbols are used for 4hr data and open symbols for 24hr data. We changed the y axis label in E to “Ratio Liver/Muscle Flux”. We aligned all axes and added in the caption “For IM administration of DLin, DODMA, DODAP, expression in liver was too low for quantification.”

12) Page 39, Figure 7, Panels D - H – Please take a similar approach for color coding and dot shapes in these plots as you do with the updated Figure 4.

We made the same color changes here as in Figure 4 above to be consistent with the reviewer’s suggestions.

Reviewer #2 (Remarks to the Author):

In their manuscript, Carrasco et al conduct a thorough characterization of various LNP compositions, including chemical, biophysical, and biological properties in order to provide some insight into LNP properties that may explain differences in levels of mRNA-mediated gene expression as well as localization of mRNA delivery following intramuscular or intravenous delivery routes.

The chemical and biophysical characterization is very well done and provides a valuable data set of interest to those in the field. I only have one major criticism with regard to the biological characterization, particularly the comparisons in gene expression levels, both in vitro and in vivo. The overall differences in expression between groups appears quite small, although presenting the data on a log scale or providing fold-change data may help the readers discern the differences more accurately. Furthermore, no statistical analyses were performed and the number of biological replicates and experimental replicates were small so at least conclusions regarding in vivo delivery efficiency cannot be drawn with any degree of certainty.

In addition to the correlation analyses that were present in the submitted manuscript, we added a multivariate linear least squares statistical analyses for the in vitro expression data in Figure 3 and the in vivo luciferase expression data in Figure 4. The in vitro data in Figure 3 was highly significant when comparing any of the LNPs resulting in $p < 0.0001$ as indicated in the revised caption. We only included the KC2 vs MC3 comparison in the figure for clarity. The in vivo data in Figure 4 was nearly significant with $p < 0.055$ comparing KC2 to MC3 for IM and IV which is quite strong considering $N=3$ in this data set. For the In vivo data of Figure 7 we retained the correlation analyses but treated the duplicates rather than the means which resulted in one of the correlations becoming significant with $p < 0.05$ for the total flux in muscle as a function of NP (blue line in Fig 7G).